# Bounded Regret for Finite-Armed Structured Bandits

**Tor Lattimore**
Department of Computing Science
University of Alberta, Canada
tlattimo@ualberta.ca

**Rémi Munos**
INRIA
Lille, France[1]
remi.munos@inria.fr

## Abstract

We study a new type of $K$-armed bandit problem where the expected return of one arm may depend on the returns of other arms. We present a new algorithm for this general class of problems and show that under certain circumstances it is possible to achieve finite expected cumulative regret. We also give problem-dependent lower bounds on the cumulative regret showing that at least in special cases the new algorithm is nearly optimal.

## 1   Introduction

The multi-armed bandit problem is a reinforcement learning problem with $K$ actions. At each time-step a learner must choose an action $i$ after which it receives a reward distributed with mean $\mu_i$. The goal is to maximise the cumulative reward. This is perhaps the simplest setting in which the well-known exploration/exploitation dilemma becomes apparent, with a learner being forced to choose between exploring arms about which she has little information, and exploiting by choosing the arm that currently appears optimal.

We consider a general class of $K$-armed bandit problems where the expected return of each arm may be dependent on other arms. This model has already been considered when the dependencies are linear [18] and also in the general setting studied here [12, 1]. Let $\Theta \ni \theta^*$ be an arbitrary

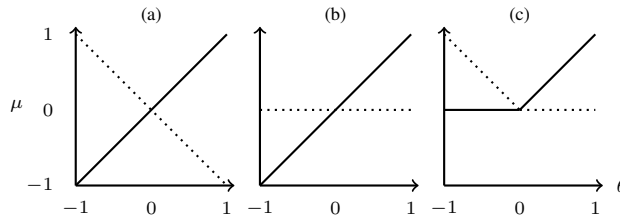

**Figure 1:** Examples

parameter space and define the expected return of arm $i$ by $\mu_i(\theta^*) \in \mathbb{R}$. The learner is permitted to know the functions $\mu_1 \cdots \mu_K$, but not the true parameter $\theta^*$. The unknown parameter $\theta^*$ determines the mean reward for each arm. The performance of a learner is measured by the (expected) cumulative regret, which is the difference between the expected return of the optimal policy and the (expected) return of the learner's policy. $R_n := n \max_{i \in 1 \cdots K} \mu_i(\theta^*) - \sum_{t=1}^{n} \mu_{I_t}(\theta^*)$ where $I_t$ is the arm chosen at time-step $t$.

A motivating example is as follows. Suppose a long-running company must decide each week whether or not to purchase some new form of advertising with unknown expected returns. The problem may be formulated using the new setting by letting $K = 2$ and $\Theta = [-\infty, \infty]$. We assume the base-line performance without purchasing the advertising is known and so define $\mu_1(\theta) = 0$ for all $\theta$. The expected return of choosing to advertise is $\mu_2(\theta) = \theta$ (see Figure (b) above).

Our main contribution is a new algorithm based on UCB [6] for the structured bandit problem with strong problem-dependent guarantees on the regret. The key improvement over UCB is that the new algorithm enjoys finite regret in many cases while UCB suffers logarithmic regret unless all arms have the same return. For example, in (a) and (c) above we show that finite regret is possible for all

$\theta^*$, while in the advertising problem finite regret is attainable if $\theta^* \geq 0$. The improved algorithm exploits the known structure and so avoids the famous negative results by Lai and Robbins [17]. One insight from this work is that knowing the return of the optimal arm and a bound on the minimum gap is not the only information that leads to the possibility of finite regret. In the examples given above neither quantity is known, but the assumed structure is nevertheless sufficient for finite regret.

Despite the enormous literature on bandits, as far as we are aware this is the first time this setting has been considered with the aim of achieving finite regret. There has been substantial work on exploiting various kinds of structure to reduce an otherwise impossible problem to one where sub-linear (or even logarithmic) regret is possible [19, 4, 10, and references therein], but the focus is usually on efficiently dealing with large action spaces rather than sub-logarithmic/finite regret. The most comparable previous work studies the case where both the return of the best arm and a bound on the minimum gap between the best arm and some sub-optimal arm is known [11, 9], which extended the permutation bandits studied by Lai and Robbins [16] and more general results by the same authors [15]. Also relevant is the paper by Agrawal et. al. [1], which studied a similar setting, but where $\Theta$ was finite. Graves and Lai [12] extended the aforementioned contribution to continuous parameter spaces (and also to MDPs). Their work differs from ours in a number of ways. Most notably, their objective is to compute exactly the asymptotically optimal regret in the case where finite regret is *not* possible. In the case where finite regret is possible they prove only that the optimal regret is sub-logarithmic, and do not present any explicit bounds on the actual regret. Aside from this the results depend on the parameter space being a metric space and they assume that the optimal policy is locally constant about the true parameter.

## 2 Notation

**General.** Most of our notation is common with [8]. The indicator function is denoted by $\mathbb{1}\{expr\}$ and is 1 if $expr$ is true and 0 otherwise. We use $\log$ for the natural logarithm. Logical and/or are denoted by $\wedge$ and $\vee$ respectively. Define function $\omega(x) = \min\{y \in \mathbb{N} : z \geq x \log z, \ \forall z \geq y\}$, which satisfies $\log \omega(x) \in O(\log x)$. In fact, $\lim_{x \to \infty} \log(\omega(x))/\log(x) = 1$.

**Bandits.** Let $\Theta$ be a set. A $K$-armed structured bandit is characterised by a set of functions $\mu_k : \Theta \to \mathbb{R}$ where $\mu_k(\theta)$ is the expected return of arm $k \in A := \{1, \cdots, K\}$ given unknown parameter $\theta$. We define the mean of the optimal arm by the function $\mu^* : \Theta \to \mathbb{R}$ with $\mu^*(\theta) := \max_i \mu_i(\theta)$. The true unknown parameter that determines the means is $\theta^* \in \Theta$. The best arm is $i^* := \arg\max_i \mu_i(\theta^*)$. The arm chosen at time-step $t$ is denoted by $I_t$ while $X_{i,s}$ is the $s$th reward obtained when sampling from arm $i$. We denote the number of times arm $i$ has been chosen at time-step $t$ by $T_i(t)$. The empiric estimate of the mean of arm $i$ based on the first $s$ samples is $\hat{\mu}_{i,s}$. We define the gap between the means of the best arm and arm $i$ by $\Delta_i := \mu^*(\theta^*) - \mu_i(\theta^*)$. The set of sub-optimal arms is $A' := \{i \in A : \Delta_i > 0\}$. The minimum gap is $\Delta_{\min} := \min_{i \in A'} \Delta_i$ while the maximum gap is $\Delta_{\max} := \max_{i \in A} \Delta_i$. The cumulative regret is defined

$$R_n := \sum_{t=1}^n \mu^*(\theta^*) - \sum_{t=1}^n \mu_{I_t} = \sum_{t=1}^n \Delta_{I_t}$$

Note quantities like $\Delta_i$ and $i^*$ depend on $\theta^*$, which is omitted from the notation. As is rather common we assume that the returns are sub-gaussian, which means that if $X$ is the return sampled from some arm, then $\ln \mathbb{E} \exp(\lambda(X - \mathbb{E}X)) \leq \lambda^2 \sigma^2/2$. As usual we assume that $\sigma^2$ is known and does not depend on the arm. If $X_1 \cdots X_n$ are sampled independently from some arm with mean $\mu$ and $S_n = \sum_{t=1}^n X_t$, then the following maximal concentration inequality is well-known.

$$\mathbb{P}\left\{\max_{1 \leq t \leq n} |S_t - t\mu| \geq \varepsilon\right\} \leq 2\exp\left(-\frac{\varepsilon^2}{2n\sigma^2}\right).$$

A straight-forward corollary is that $\mathbb{P}\{|\hat{\mu}_{i,n} - \mu_i| \geq \varepsilon\} \leq 2\exp\left(-\frac{\varepsilon^2 n}{2\sigma^2}\right)$.

It is an important point that $\Theta$ is completely arbitrary. The classic multi-armed bandit can be obtained by setting $\Theta = \mathbb{R}^K$ and $\mu_k(\theta) = \theta_k$, which removes all dependencies between the arms. The setting where the optimal expected return is known to be zero and a bound on $\Delta_i \geq \varepsilon$ is known can be regained by choosing $\Theta = (-\infty, -\varepsilon]^K \times \{1, \cdots, K\}$ and $\mu_k(\theta_1, \cdots, \theta_K, i) = \theta_k \mathbb{1}\{k \neq i\}$. We do not demand that $\mu_k : \Theta \to \mathbb{R}$ be continuous, or even that $\Theta$ be endowed with a topology.

# 3 Structured UCB

We propose a new algorithm called UCB-S that is a straight-forward modification of UCB [6], but where the known structure of the problem is exploited. At each time-step it constructs a confidence interval about the mean of each arm. From this a subspace $\tilde{\Theta}_t \subseteq \Theta$ is constructed, which contains the true parameter $\theta$ with high probability. The algorithm takes the optimistic action over all $\theta \in \tilde{\Theta}_t$.

---

**Algorithm 1** UCB-S

---

1: **Input:** functions $\mu_1, \cdots, \mu_k : \Theta \to [0,1]$
2: **for** $t \in 1, \ldots, \infty$ **do**

3:     Define confidence set $\tilde{\Theta}_t \leftarrow \left\{ \tilde{\theta} : \forall i, \ \left| \mu_i(\tilde{\theta}) - \hat{\mu}_{i,T_i(t-1)} \right| < \sqrt{\dfrac{\alpha \sigma^2 \log t}{T_i(t-1)}} \right\}$

4:     **if** $\tilde{\Theta}_t = \emptyset$ **then**
5:         Choose arm arbitrarily
6:     **else**
7:         Optimistic arm is $i \leftarrow \arg\max_i \sup_{\tilde{\theta} \in \tilde{\Theta}_t} \mu_i(\tilde{\theta})$
8:         Choose arm $i$

---

**Remark 1.** The choice of arm when $\tilde{\Theta}_t = \emptyset$ does not affect the regret bounds in this paper. In practice, it is possible to simply increase $t$ without taking an action, but this complicates the analysis. In many cases the true parameter $\theta^*$ is never identified in the sense that we do not expect that $\tilde{\Theta}_t \to \{\theta^*\}$. The computational complexity of UCB-S depends on the difficulty of computing $\tilde{\Theta}_t$ and computing the optimistic arm within this set. This is efficient in simple cases, like when $\mu_k$ is piecewise linear, but may be intractable for complex functions.

## 4 Theorems

We present two main theorems bounding the regret of the UCB-S algorithm. The first is for arbitrary $\theta^*$, which leads to a logarithmic bound on the regret comparable to that obtained for UCB by [6]. The analysis is slightly different because UCB-S maintains upper and lower confidence bounds and selects its actions optimistically from the model class, rather than by maximising the upper confidence bound as UCB does.

**Theorem 2.** *If $\alpha > 2$ and $\theta \in \Theta$, then the algorithm UCB-S suffers an expected regret of at most*

$$\mathbb{E}R_n \leq \frac{2\Delta_{\max} K(\alpha - 1)}{\alpha - 2} + \sum_{i \in A'} \frac{8\alpha\sigma^2 \log n}{\Delta_i} + \sum_i \Delta_i$$

If the samples from the optimal arm are sufficient to learn the optimal action, then finite regret is possible. In Section 6 we give something of a converse by showing that if knowing the mean of the optimal arm is insufficient to act optimally, then logarithmic regret is unavoidable.

**Theorem 3.** *Let $\alpha = 4$ and assume there exists an $\varepsilon > 0$ such that*

$$(\forall \theta \in \Theta) \qquad |\mu_{i^*}(\theta^*) - \mu_{i^*}(\theta)| < \varepsilon \implies \forall i \neq i^*, \mu_{i^*}(\theta) > \mu_i(\theta). \tag{1}$$

*Then* $\mathbb{E}R_n \leq \sum_{i \in A'} \left( \dfrac{32\sigma^2 \log \omega^*}{\Delta_i} + \Delta_i \right) + 3\Delta_{\max} K + \dfrac{\Delta_{\max} K^3}{\omega^*},$
*with* $\omega^* := \max\left\{ \omega\left(\dfrac{8\sigma^2 \alpha K}{\varepsilon^2}\right), \ \omega\left(\dfrac{8\sigma^2 \alpha K}{\Delta_{\min}^2}\right) \right\}.$

**Remark 4.** For small $\varepsilon$ and large $n$ the expected regret looks like $\mathbb{E}R_n \in O\left(\sum_{i=1}^K \dfrac{\log\left(\frac{1}{\varepsilon}\right)}{\Delta_i}\right)$ (for small $n$ the regret is, of course, even smaller).

The explanation of the bound is as follows. If at some time-step $t$ it holds that all confidence intervals contain the truth and the width of the confidence interval about $i^*$ drops below $\varepsilon$, then by the condition in Equation (1) it holds that $i^*$ is the optimistic arm within $\tilde{\Theta}_t$. In this case UCB-S

suffers no regret at this time-step. Since the number of samples of each sub-optimal arm grows at most logarithmically by the proof of Theorem 2, the number of samples of the best arm must grow linearly. Therefore the number of time-steps before best arm has been pulled $O(\varepsilon^{-2})$ times is also $O(\varepsilon^{-2})$. After this point the algorithm suffers only a constant cumulative penalty for the possibility that the confidence intervals do not contain the truth, which is finite for suitably chosen values of $\alpha$. Note that Agrawal et. al. [1] had essentially the same condition to achieve finite regret as (1), but specified to the case where $\Theta$ is finite.

An interesting question is raised by comparing the bound in Theorem 3 to those given by Bubeck et. al. [11] where if the expected return of the best arm is known and $\varepsilon$ is a known bound on the minimum gap, then a regret bound of

$$O\left(\sum_{i \in A'}\left(\frac{\log\left(\frac{2\Delta_i}{\varepsilon}\right)}{\Delta_i}\left(1 + \log\log\frac{1}{\varepsilon}\right)\right)\right) \tag{2}$$

is achieved. If $\varepsilon$ is close to $\Delta_i$, then this bound is an improvement over the bound given by Theorem 3, although our theorem is more general. The improved UCB algorithm [7] enjoys a bound on the expected regret of $O(\sum_{i \in A'} \frac{1}{\Delta_i} \log n \Delta_i^2)$. If we follow the same reasoning as above we obtain a bound comparable to (2). Unfortunately though, the extension of the improved UCB algorithm to the structured setting is rather challenging with the main obstruction being the extreme growth of the phases used by improved UCB. Refining the phases leads to super-logarithmic regret, a problem we ultimately failed to resolve. Nevertheless we feel that there is some hope of obtaining a bound like (2) in this setting.

Before the proofs of Theorems 2 and 3 we give some example structured bandits and indicate the regions where the conditions for Theorem 3 are (not) met. Areas where Theorem 3 can be applied to obtain finite regret are unshaded while those with logarithmic regret are shaded.

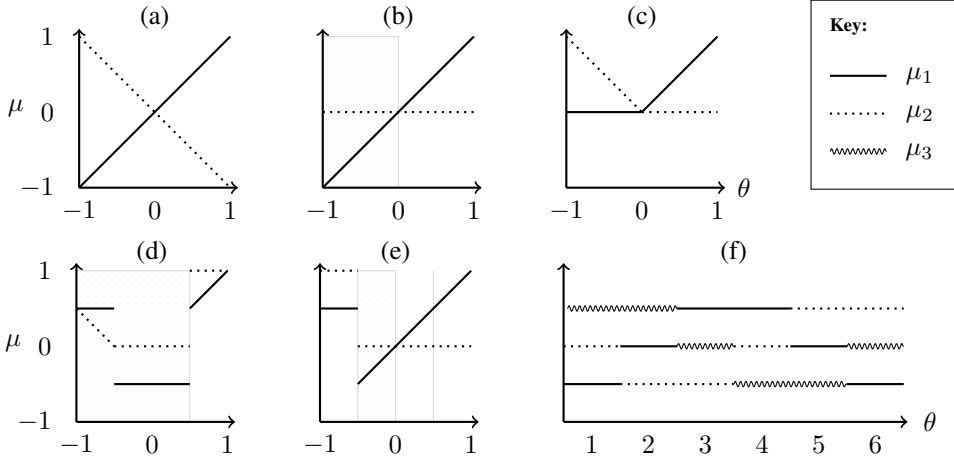

**Figure 2:** Examples

(a) The conditions for Theorem 3 are met for all $\theta \neq 0$, but for $\theta = 0$ the regret strictly vanishes for all policies, which means that the regret is bounded by $\mathbb{E}R_n \in O(\mathbb{1}\{\theta^* \neq 0\}\frac{1}{|\theta^*|}\log\frac{1}{|\theta^*|})$.

(b) Action 2 is uninformative and not globally optimal so Theorem 3 does not apply for $\theta < 1/2$ where this action is optimal. For $\theta > 0$ the optimal action is 1, when the conditions are met and finite regret is again achieved.

$$\mathbb{E}R_n \in O\left(\mathbb{1}\{\theta^* < 0\}\frac{\log n}{|\theta^*|} + \mathbb{1}\{\theta^* > 0\}\frac{\log\frac{1}{\theta^*}}{\theta^*}\right).$$

(c) The conditions for Theorem 3 are again met for all non-zero $\theta^*$, which leads as in (a) to a regret of $\mathbb{E}R_n \in O(\mathbb{1}\{\theta^* \neq 0\}\frac{1}{|\theta^*|}\log\frac{1}{|\theta^*|})$.

Examples (d) and (e) illustrate the potential complexity of the regions in which finite regret is possible. Note especially that in (e) the regret for $\theta^* = \frac{1}{2}$ is logarithmic in the horizon, but finite for $\theta^*$ arbitrarily close. Example (f) is a permutation bandit with 3 arms where it can be clearly seen that the conditions of Theorem 3 are satisfied.

# 5 Proof of Theorems 2 and 3

We start by bounding the probability that some mean does not lie inside the confidence set.

**Lemma 5.** $\mathbb{P}\{F_t = 1\} \leq 2Kt \exp(-\alpha \log(t))$ *where*

$$F_t = \mathbb{1}\left\{\exists i : |\hat{\mu}_{i,T_i(t-1)} - \mu_i| \geq \sqrt{\frac{2\alpha\sigma^2 \log t}{T_i(t-1)}}\right\}.$$

*Proof.* We use the concentration guarantees:

$$\mathbb{P}\{F_t = 1\} \overset{(a)}{=} \mathbb{P}\left\{\exists i : |\mu_i(\theta^*) - \hat{\mu}_{i,T_i(t-1)}| \geq \sqrt{\frac{2\alpha\sigma^2 \log t}{T_i(t-1)}}\right\}$$

$$\overset{(b)}{\leq} \sum_{i=1}^{K} \mathbb{P}\left\{|\mu_i(\theta^*) - \hat{\mu}_{i,T_i(t-1)}| \geq \sqrt{\frac{2\alpha\sigma^2 \log t}{T_i(t-1)}}\right\}$$

$$\overset{(c)}{\leq} \sum_{i=1}^{K}\sum_{s=1}^{t} \mathbb{P}\left\{|\mu_i(\theta^*) - \hat{\mu}_{i,s}| \geq \sqrt{\frac{2\alpha\sigma^2 \log t}{s}}\right\} \overset{(d)}{\leq} \sum_{i=1}^{K}\sum_{s=1}^{t} 2\exp(-\alpha \log t) \overset{(e)}{=} 2Kt^{1-\alpha}$$

where (a) follows from the definition of $F_t$. (b) by the union bound. (c) also follows from the union bound and is the standard trick to deal with the random variable $T_i(t-1)$. (d) follows from the concentration inequalities for sub-gaussian random variables. (e) is trivial. $\square$

*Proof of Theorem 2.* Let $i$ be an arm with $\Delta_i > 0$ and suppose that $I_t = i$. Then either $F_t$ is true or

$$T_i(t-1) < \left\lceil \frac{8\sigma^2\alpha \log n}{\Delta_i^2} \right\rceil =: u_i(n) \tag{3}$$

Note that if $F_t$ does not hold then the true parameter lies within the confidence set, $\theta^* \in \tilde{\Theta}_t$. Suppose on the contrary that $F_t$ and (3) are both false.

$$\max_{\tilde{\theta}\in\tilde{\Theta}_t} \mu_{i^*}(\tilde{\theta}) \overset{(a)}{\geq} \mu^*(\theta^*) \overset{(b)}{=} \mu_i(\theta^*) + \Delta_i \overset{(c)}{>} \Delta_i + \hat{\mu}_{i,T_i(t-1)} - \sqrt{\frac{2\sigma^2\alpha \log t}{T_i(t-1)}}$$

$$\overset{(d)}{\geq} \hat{\mu}_{i,T_i(t-1)} + \sqrt{\frac{2\alpha\sigma^2 \log t}{T_i(t-1)}} \overset{(e)}{\geq} \max_{\tilde{\theta}\in\tilde{\Theta}_t} \mu_i(\tilde{\theta}),$$

where (a) follows since $\theta^* \in \tilde{\Theta}_t$. (b) is the definition of the gap. (c) since $F_t$ is false. (d) is true because (3) is false. Therefore arm $i$ is not taken. We now bound the expected number of times that arm $i$ is played within the first $n$ time-steps by

$$\mathbb{E}T_i(n) \overset{(a)}{=} \mathbb{E}\sum_{t=1}^{n} \mathbb{1}\{I_t = i\} \overset{(b)}{\leq} u_i(n) + \mathbb{E}\sum_{t=u_i+1}^{n} \mathbb{1}\{I_t = i \wedge (3) \text{ is false}\}$$

$$\overset{(c)}{\leq} u_i(n) + \mathbb{E}\sum_{t=u_i+1}^{n} \mathbb{1}\{F_t = 1 \wedge I_t = i\}$$

where (a) follows from the linearity of expectation and definition of $T_i(n)$. (b) by Equation (3) and the definition of $u_i(n)$ and expectation. (c) is true by recalling that playing arm $i$ at time-step $t$ implies that either $F_t$ or (3) must be true. Therefore

$$\mathbb{E}R_n \leq \sum_{i\in A'} \Delta_i \left(u_i(n) + \mathbb{E}\sum_{t=u_i+1}^{n} \mathbb{1}\{F_t = 1 \wedge I_t = i\}\right) \leq \sum_{i\in A'} \Delta_i u_i(n) + \Delta_{\max}\mathbb{E}\sum_{t=1}^{n} \mathbb{1}\{F_t = 1\} \tag{4}$$

Bounding the second summation

$$\mathbb{E}\sum_{t=1}^{n} \mathbb{1}\{F_t = 1\} \overset{(a)}{=} \sum_{t=1}^{n} \mathbb{P}\{F_t = 1\} \overset{(b)}{\leq} \sum_{t=1}^{n} 2Kt^{1-\alpha} \overset{(c)}{\leq} \frac{2K(\alpha-1)}{\alpha-2}$$

where (a) follows by exchanging the expectation and sum and because the expectation of an indicator function can be written as the probability of the event. (b) by Lemma 5 and (c) is trivial. Substituting into (4) leads to

$$\mathbb{E}R_n \leq \frac{2\Delta_{\max}K(\alpha-1)}{\alpha-2} + \sum_{i\in A'}\frac{8\alpha\sigma^2\log n}{\Delta_i} + \sum_i \Delta_i. \qquad \square$$

Before the proof of Theorem 3 we need a high-probability bound on the number of times arm $i$ is pulled, which is proven along the lines of similar results by [5].

**Lemma 6.** *Let $i \in A'$ be some sub-optimal arm. If $z > u_i(n)$, then $\mathbb{P}\{T_i(n) > z\} \leq \frac{2Kz^{2-\alpha}}{\alpha-2}$.*

*Proof.* As in the proof of Theorem 2, if $t \leq n$ and $F_t$ is false and $T_i(t-1) > u_i(n) \geq u_i(t)$, then arm $i$ is not chosen. Therefore

$$\mathbb{P}\{T_i(n) > z\} \leq \sum_{t=z+1}^{n} \mathbb{P}\{F_t = 1\} \overset{(a)}{\leq} \sum_{t=z+1}^{n} 2Kt^{1-\alpha} \overset{(b)}{\leq} 2K\int_z^n t^{1-\alpha}dt \overset{(c)}{\leq} \frac{2Kz^{2-\alpha}}{\alpha-2}$$

where (a) follows from Lemma 5 and (b) and (c) are trivial. $\qquad \square$

**Lemma 7.** *Assume the conditions of Theorem 3 and additionally that $T_{i^*}(t-1) \geq \left\lceil \frac{8\alpha\sigma^2\log t}{\varepsilon^2} \right\rceil$ and $F_t$ is false. Then $I_t = i^*$.*

*Proof.* Since $F_t$ is false, for $\tilde{\theta} \in \tilde{\Theta}_t$ we have:

$$|\mu_{i^*}(\tilde{\theta}) - \mu_{i^*}(\theta^*)| \overset{(a)}{\leq} |\mu_{i^*}(\tilde{\theta}) - \hat{\mu}_{i^*,T_i(t-1)}| + |\hat{\mu}_{i^*,T_i(t-1)} - \mu_{i^*}(\theta^*)| \overset{(b)}{<} 2\sqrt{\frac{2\sigma^2\alpha\log t}{T_{i^*}(t-1)}} \overset{(c)}{\leq} \varepsilon$$

where (a) is the triangle inequality. (b) follows by the definition of the confidence interval and because $F_t$ is false. (c) by the assumed lower bound on $T_{i^*}(t-1)$. Therefore by (1), for all $\tilde{\theta} \in \tilde{\Theta}_t$ it holds that the best arm is $i^*$. Finally, since $F_t$ is false, $\theta^* \in \tilde{\Theta}_t$, which means that $\tilde{\Theta}_t \neq \emptyset$. Therefore $I_t = i^*$ as required. $\qquad \square$

*Proof of Theorem 3.* Let $\omega^*$ be some constant to be chosen later. Then the regret may be written as

$$\mathbb{E}R_n \leq \mathbb{E}\sum_{t=1}^{\omega^*}\sum_{i=1}^{K}\Delta_i\mathbb{1}\{I_t = i\} + \Delta_{\max}\mathbb{E}\sum_{t=\omega^*+1}^{n}\mathbb{1}\{I_t \neq i^*\}. \qquad (5)$$

The first summation is bounded as in the proof of Theorem 2 by

$$\mathbb{E}\sum_{t=1}^{\omega^*}\sum_{i\in A}\Delta_i\mathbb{1}\{I_t = i\} \leq \sum_{i\in A'}\left(\Delta_i + \frac{8\alpha\sigma^2\log\omega^*}{\Delta_i}\right) + \sum_{t=1}^{\omega^*}\mathbb{P}\{F_t = 1\}. \qquad (6)$$

We now bound the second sum in (5) and choose $\omega^*$. By Lemma 6, if $\frac{n}{K} > u_i(n)$, then

$$\mathbb{P}\left\{T_i(n) > \frac{n}{K}\right\} \leq \frac{2K}{\alpha-2}\left(\frac{K}{n}\right)^{\alpha-2}. \qquad (7)$$

Suppose $t \geq \omega^* := \max\left\{\omega\left(\frac{8\sigma^2\alpha K}{\varepsilon^2}\right), \omega\left(\frac{8\sigma^2\alpha K}{\Delta_{\min}^2}\right)\right\}$. Then $\frac{t}{K} > u_i(t)$ for all $i \neq i^*$ and $\frac{t}{K} \geq \frac{8\sigma^2\alpha\log t}{\varepsilon^2}$. By the union bound

$$\mathbb{P}\left\{T_{i^*}(t) < \frac{8\sigma^2\alpha\log t}{\varepsilon^2}\right\} \overset{(a)}{\leq} \mathbb{P}\left\{T_{i^*}(t) < \frac{t}{K}\right\} \overset{(b)}{\leq} \mathbb{P}\left\{\exists i : T_i(t) > \frac{t}{K}\right\} \overset{(c)}{<} \frac{2K^2}{\alpha-2}\left(\frac{K}{t}\right)^{\alpha-2}$$

$$(8)$$

where (a) is true since $\frac{t}{K} \geq \frac{8\sigma^2\alpha\log t}{\varepsilon^2}$. (b) since $\sum_{i=1}^K T_i(t) = t$. (c) by the union bound and (7). Now if $T_i(t) \geq \frac{8\sigma^2\alpha\log t}{\varepsilon^2}$ and $F_t$ is false, then the chosen arm is $i^*$. Therefore

$$
\begin{aligned}
\mathbb{E}\sum_{t=\omega^*+1}^n \mathbb{1}\{I_t \neq i^*\} &\leq \sum_{t=\omega^*+1}^n \mathbb{P}\{F_t=1\} + \sum_{t=\omega^*+1}^n \mathbb{P}\left\{T_i(t-1) < \frac{8\sigma^2\alpha\log t}{\varepsilon^2}\right\} \\
&\overset{(a)}{\leq} \sum_{t=\omega^*+1}^n \mathbb{P}\{F_t=1\} + \frac{2K^2}{\alpha-2}\sum_{t=\omega^*+1}^n \left(\frac{K}{t}\right)^{\alpha-2} \\
&\overset{(b)}{\leq} \sum_{t=\omega^*+1}^n \mathbb{P}\{F_t=1\} + \frac{2K^2}{(\alpha-2)(\alpha-3)}\left(\frac{K}{\omega^*}\right)^{\alpha-3}
\end{aligned}
$$

(9)

where (a) follows from (8) and (b) by straight-forward calculus. Therefore by combining (5), (6) and (9) we obtain

$$
\begin{aligned}
\mathbb{E}R_n &\leq \sum_{i:\Delta_i>0}\Delta_i\left\lceil\frac{8\sigma^2\alpha\log\omega^*}{\Delta_i^2}\right\rceil + \frac{2\Delta_{\max}K^2}{(\alpha-2)(\alpha-3)}\left(\frac{K}{\omega^*}\right)^{\alpha-3} + \Delta_{\max}\sum_{t=1}^n \mathbb{P}\{F_t=1\} \\
&\leq \sum_{i:\Delta_i>0}\Delta_i\left\lceil\frac{8\sigma^2\alpha\log\omega^*}{\Delta_i^2}\right\rceil + \frac{2\Delta_{\max}K^2}{(\alpha-2)(\alpha-3)}\left(\frac{K}{\omega^*}\right)^{\alpha-3} + \frac{2\Delta_{\max}K(\alpha-1)}{\alpha-2}
\end{aligned}
$$

Setting $\alpha=4$ leads to $\mathbb{E}R_n \leq \sum_{i=1}^K\left(\frac{32\sigma^2\log\omega^*}{\Delta_i} + \Delta_i\right) + 3\Delta_{\max}K + \frac{\Delta_{\max}K^3}{\omega^*}$. $\qquad\square$

## 6 Lower Bounds and Ambiguous Examples

We prove lower bounds for two illustrative examples of structured bandits. Some previous work is also relevant. The famous paper by Lai and Robbins [17] shows that the bound of Theorem 2 cannot in general be greatly improved. Many of the techniques here are borrowed from Bubeck et. al. [11]. Given a fixed algorithm and varying $\theta$ we denote the regret and expectation by $R_n(\theta)$ and $\mathbb{E}_\theta$ respectively. Returns are assumed to be sampled from a normal distribution with unit variance, so that $\sigma^2 = 1$. The proofs of the following theorems may be found in the supplementary material.

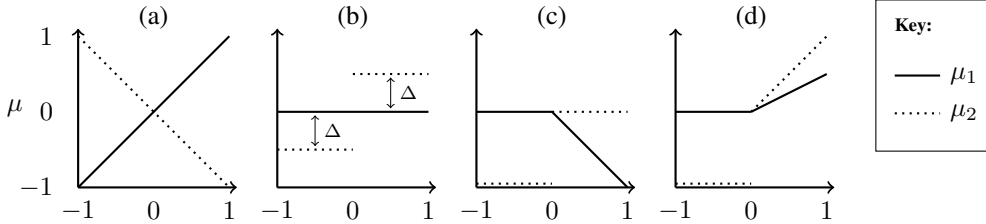

**Figure 3:** Counter-examples

**Theorem 8.** *Given the structured bandit depicted in Figure 3.(a) or Figure 2.(c), then for all $\theta > 0$ and all algorithms the regret satisfies $\max\{\mathbb{E}_{-\theta}R_n(-\theta), \mathbb{E}_\theta R_n(\theta)\} \geq \frac{1}{8\theta}$ for sufficiently large $n$.*

**Theorem 9.** *Let $\Theta, \{\mu_1, \mu_2\}$ be a structured bandit where returns are sampled from a normal distribution with unit variance. Assume there exists a pair $\theta_1, \theta_2 \in \Theta$ and constant $\Delta > 0$ such that $\mu_1(\theta_1) = \mu_1(\theta_2)$ and $\mu_1(\theta_1) \geq \mu_2(\theta_1) + \Delta$ and $\mu_2(\theta_2) \geq \mu_1(\theta_2) + \Delta$. Then the following hold:*

*(1)* $\mathbb{E}_{\theta_1}R_n(\theta_1) \geq \frac{1+\log 2n\Delta^2}{8\Delta} - \frac{1}{2}\mathbb{E}_{\theta_2}R_n(\theta_2)$

*(2)* $\mathbb{E}_{\theta_2}R_n(\theta_2) \geq \frac{n\Delta}{2}\exp\left(-4\mathbb{E}_{\theta_1}R_n(\theta_1)\Delta\right) - \mathbb{E}_{\theta_1}R_n(\theta_1)$

A natural example where the conditions are satisfied is depicted in Figure 3.(b) and by choosing $\theta_1 = -1, \theta_2 = 1$. We know from Theorem 3 that UCB-S enjoys finite regret of $\mathbb{E}_{\theta_2}R_n(\theta_2) \in O(\frac{1}{\Delta}\log\frac{1}{\Delta})$ and logarithmic regret $\mathbb{E}_{\theta_1}R_n(\theta_1) \in O(\frac{1}{\Delta}\log n)$. Part 1 of Theorem 9 shows that if we demand finite regret $\mathbb{E}_{\theta_2}R_n(\theta_2) \in O(1)$, then the regret $\mathbb{E}_{\theta_1}R_n(\theta_1)$ is necessarily logarithmic. On the other

hand, part 2 shows that if we demand $\mathbb{E}_{\theta_1} R_n(\theta_1) \in o(\log(n))$, then the regret $\mathbb{E}_{\theta_2} R_n(\theta_2) \in \Omega(n)$. Therefore the trade-off made by UCB-S essentially cannot be improved.

**Discussion of Figure 3.(c/d).** In both examples there is an ambiguous region for which the lower bound (Theorem 9) does not show that logarithmic regret is unavoidable, but where Theorem 3 cannot be applied to show that UCB-S achieves finite regret. We managed to show that finite regret is possible in both cases by using a different algorithm. For (c) we could construct a carefully tuned algorithm for which the regret was at most $O(1)$ if $\theta \leq 0$ and $O(\frac{1}{\theta} \log \log \frac{1}{\theta})$ otherwise. This result contradicts a claim by Bubeck et. al. [11, Thm. 8]. Additional discussion of the ambiguous case in general, as well as this specific example, may be found in the supplementary material. One observation is that unbridled optimism is the cause of the failure of UCB-S in these cases. This is illustrated by Figure 3.(d) with $\theta \leq 0$. No matter how narrow the confidence interval about $\mu_1$, if the second action has not been taken sufficiently often, then there will still be some belief that $\theta > 0$ is possible where the second action is optimistic, which leads to logarithmic regret. Adapting the algorithm to be slightly risk averse solves this problem.

## 7 Experiments

We tested Algorithm 1 on a selection of structured bandits depicted in Figure 2 and compared to UCB [6, 8]. Rewards were sampled from normal distributions with unit variances. For UCB we chose $\alpha = 2$, while we used the theoretically justified $\alpha = 4$ for Algorithm 1. All code is available in the supplementary material. Each data-point is the average of 500 independent samples with the blue crosses and red squares indicating the regret of UCB-S and UCB respectively.

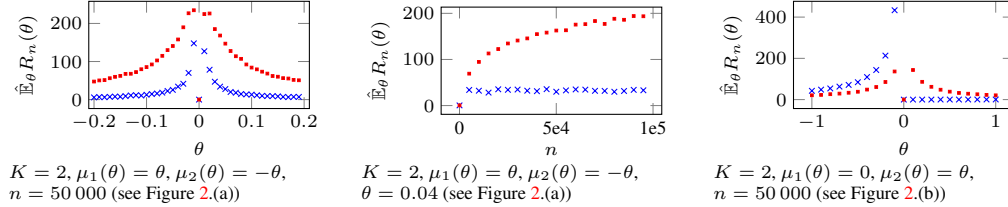

$K = 2, \mu_1(\theta) = \theta, \mu_2(\theta) = -\theta,$
$n = 50\,000$ (see Figure 2.(a))

$K = 2, \mu_1(\theta) = \theta, \mu_2(\theta) = -\theta,$
$\theta = 0.04$ (see Figure 2.(a))

$K = 2, \mu_1(\theta) = 0, \mu_2(\theta) = \theta,$
$n = 50\,000$ (see Figure 2.(b))

The results show that Algorithm 1 typically out-performs regular UCB. The exception is the top right experiment where UCB performs slightly better for $\theta < 0$. This is not surprising, since in this case the structured version of UCB cannot exploit the additional structure and suffers due to worse constant factors. On the other hand, if $\theta > 0$, then UCB endures logarithmic regret and performs significantly worse than its structured counterpart. The superiority of Algorithm 1 would be accentuated in the top left and bottom right experiments by increasing the horizon.

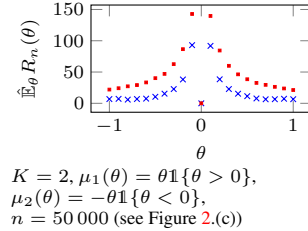

$K = 2, \mu_1(\theta) = \theta \mathbb{1}\{\theta > 0\},$
$\mu_2(\theta) = -\theta \mathbb{1}\{\theta < 0\},$
$n = 50\,000$ (see Figure 2.(c))

## 8 Conclusion

The limitation of the new approach is that the proof techniques and algorithm are most suited to the case where the number of actions is relatively small. Generalising the techniques to large action spaces is therefore an important open problem. There is still a small gap between the upper and lower bounds, and the lower bounds have only been proven for special examples. Proving a general problem-dependent lower bound is an interesting question, but probably extremely challenging given the flexibility of the setting. We are also curious to know if there exist problems for which the optimal regret is somewhere between finite and logarithmic. Another question is that of how to define Thompson sampling for structured bandits. Thompson sampling has recently attracted a great deal of attention [13, 2, 14, 3, 9], but so far we are unable even to define an algorithm resembling Thompson sampling for the general structured bandit problem.

**Acknowledgements.** Tor Lattimore was supported by the Google Australia Fellowship for Machine Learning and the Alberta Innovates Technology Futures, NSERC. The majority of this work was completed while Rémi Munos was visiting Microsoft Research, New England. This research was partially supported by the European Community's Seventh Framework Programme under grant agreements no. 270327 (project CompLACS).

## Footnotes

[1]Current affiliation: Google DeepMind.

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
