[Supplementary Material]

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

*Proof.* The proof uses the same technique as the proof of Theorem 5 in the paper by [11]. Fix an algorithm and let $\mathbb{P}_{\theta,t}$ be the probability measure on the space of outcomes up to time-step $t$ under the bandit determined by parameter $\theta$.

$$
\begin{aligned}
\mathbb{E}_{-\theta} R_n(-\theta) + \mathbb{E}_\theta R_n(\theta) &\overset{(a)}{=} 2\theta \left( \mathbb{E}_{-\theta} \sum_{t=1}^{n} \mathbb{1}\{I_t = 1\} + \mathbb{E}_\theta \sum_{t=1}^{n} \mathbb{1}\{I_t = 2\} \right) \\
&\overset{(b)}{=} 2\theta \sum_{t=1}^{n} \left( \mathbb{P}_{-\theta,t}\{I_t = 1\} + \mathbb{P}_{\theta,t}\{I_t = 2\} \right) \overset{(c)}{\geq} \theta \sum_{t=1}^{n} \exp\left( -\mathrm{KL}(\mathbb{P}_{-\theta,t}, \mathbb{P}_{\theta,t}) \right) \\
&\overset{(d)}{=} \theta \sum_{t=1}^{n} \exp\left( -4t\theta^2 \right) \overset{(e)}{\geq} \frac{1}{8\theta}
\end{aligned}
$$

where (a) follows since $2|\theta|$ is the gap between the expected returns of the two arms given parameter $\theta$ and by the definition of the regret. (b) by replacing the expectations with probabilities. (c) follows from Lemma 4 by [11] where $\mathrm{KL}(\mathbb{P}_{-\theta,t}, \mathbb{P}_{\theta,t})$ is the relative entropy between measures $\mathbb{P}_{-\theta,t}$ and $\mathbb{P}_{\theta,t}$. (d) is true by computing the relative entropy between two normals with unit variance and means separated by $2\theta$, which is $4\theta^2$. (e) holds for sufficiently large $n$. $\qquad\square$

**Theorem 9.** *Let $\Theta, \{\mu_1, \mu_2\}$ be a structured bandit where returns are sampled from a normal distribution with unit variance. Assume there exists a pair $\theta_1, \theta_2 \in \Theta$ and constant $\Delta > 0$ such that $\mu_1(\theta_1) = \mu_1(\theta_2)$ and $\mu_1(\theta_1) \geq \mu_2(\theta_1) + \Delta$ and $\mu_2(\theta_2) \geq \mu_1(\theta_2) + \Delta$. Then the following hold:*

*(1) $\mathbb{E}_{\theta_1} R_n(\theta_1) \geq \frac{1 + \log 2n\Delta^2}{8\Delta} - \frac{1}{2}\mathbb{E}_{\theta_2} R_n(\theta_2)$*

*(2) $\mathbb{E}_{\theta_2} R_n(\theta_2) \geq \frac{n\Delta}{2} \exp\left(-4\mathbb{E}_{\theta_1} R_n(\theta_1)\Delta\right) - \mathbb{E}_{\theta_1} R_n(\theta_1)$*

A natural example where the conditions are satisfied is depicted in Figure 3.(b) and by choosing $\theta_1 = -1, \theta_2 = 1$. We know from Theorem 3 that UCB-S enjoys finite regret of $\mathbb{E}_{\theta_2} R_n(\theta_2) \in O(\frac{1}{\Delta}\log\frac{1}{\Delta})$ and logarithmic regret $\mathbb{E}_{\theta_1} R_n(\theta_1) \in O(\frac{1}{\Delta}\log n)$. Part 1 of Theorem 9 shows that if we demand finite regret $\mathbb{E}_{\theta_2} R_n(\theta_2) \in O(1)$, then the regret $\mathbb{E}_{\theta_1} R_n(\theta_1)$ is necessarily logarithmic. On the other hand, part 2 shows that if we demand $\mathbb{E}_{\theta_1} R_n(\theta_1) \in o(\log(n))$, then the regret $\mathbb{E}_{\theta_2} R_n(\theta_2) \in \Omega(n)$. Therefore the trade-off made by UCB-S essentially cannot be improved.

*Proof of Theorem 9.* Again, we make use of the techniques of [11].

$$
\mathbb{E}_{\theta_1} R_n(\theta_1) + \mathbb{E}_{\theta_2} R_n(\theta_2) \overset{(a)}{\geq} \Delta\left(\mathbb{E}_{\theta_1} T_2(n) + \mathbb{E}_{\theta_2} T_1(n)\right) \overset{(b)}{\geq} \Delta \sum_{t=1}^{n} \left(\mathbb{P}_{\theta_1}\{I_t = 2\} + \mathbb{P}_{\theta_2}\{I_t = 1\}\right)
$$

$$
\overset{(c)}{\geq} \frac{\Delta}{2} \sum_{t=1}^{n} \exp\left(-\mathrm{KL}(\mathbb{P}_{\theta_1,t}, \mathbb{P}_{\theta_2,t})\right) \overset{(d)}{\geq} \frac{n\Delta}{2}\exp\left(-\mathrm{KL}(\mathbb{P}_{\theta_1,n}, \mathbb{P}_{\theta_2,n})\right)
$$

$$
\overset{(e)}{\geq} \frac{n\Delta}{2}\exp\left(-4\Delta^2 \mathbb{E}_{\theta_1} T_2(n)\right) \overset{(f)}{\geq} \frac{n\Delta}{2}\exp\left(-4\Delta\mathbb{E}_{\theta_1} R_n(\theta_1)\right) \qquad (10)
$$

where (a) follows from the definition of the regret and the bandits used. (b) by the definition of $T_k(n)$. (c) by Lemma 4 of [11]. (d) since the relative entropy $\mathrm{KL}(\mathbb{P}_{\theta_1,t}, \mathbb{P}_{\theta_2,t})$ is increasing with $t$. (e) By checking that $\mathrm{KL}(\mathbb{P}_{\theta_1,n}, \mathbb{P}_{\theta_2,n}) = 4\Delta^2\mathbb{E}_{\theta_1} T_2(n)$. (f) by substituting the definition of the regret. Now part 2 is completed by rearranging (10). For part 1 we also rearrange (10) to obtain

$$
\mathbb{E}_{\theta_1} R_n(\theta_1) \geq \frac{n\Delta}{2}\exp\left(-4\Delta\mathbb{E}_{\theta_1} R_n(\theta_1)\right) - \mathbb{E}_{\theta_2} R_n(\theta_2)
$$

Letting $x = \mathbb{E}_{\theta_1} R_n(\theta_1)$ and using the constraint above we obtain:

$$
x \geq \frac{x}{2} + \frac{1}{2}\left(\frac{n\Delta}{2}\exp\left(-4\Delta x\right) - \mathbb{E}_{\theta_2} R_n(\theta_2)\right).
$$

But by simple calculus the function on the right hand side is minimised for $x = \frac{1}{4\Delta}\log(2n\Delta^2)$, which leads to

$$
\mathbb{E}_{\theta_1} R_n(\theta_1) \geq \frac{\log(2n\Delta^2)}{8\Delta} + \frac{1}{8\Delta} - \frac{1}{2}\mathbb{E}_{\theta_2} R_n(\theta_2). \qquad\square
$$

**Discussion of Figure 3.(c/d).** In both examples there is an ambiguous region for which the lower bound (Theorem 9) does not show that logarithmic regret is unavoidable, but where Theorem 3 cannot be applied to show that UCB-S achieves finite regret. We managed to show that finite regret is possible in both cases by using a different algorithm. For (c) we could construct a carefully tuned algorithm for which the regret was at most $O(1)$ if $\theta \leq 0$ and $O(\frac{1}{\theta}\log\log\frac{1}{\theta})$ otherwise. This result contradicts a claim by Bubeck et. al. [11, Thm. 8]. Additional discussion of the ambiguous case in general, as well as this specific example, may be found in the supplementary material. One observation is that unbridled optimism is the cause of the failure of UCB-S in these cases. This is illustrated by Figure 3.(d) with $\theta \leq 0$. No matter how narrow the confidence interval about $\mu_1$, if the second action has not been taken sufficiently often, then there will still be some belief that $\theta > 0$ is possible where the second action is optimistic, which leads to logarithmic regret. Adapting the algorithm to be slightly risk averse solves this problem.

# 7 Experiments

We tested Algorithm 1 on a selection of structured bandits depicted in Figure 2 and compared to UCB [6, 8]. Rewards were sampled from normal distributions with unit variances. For UCB we chose $\alpha = 2$, while we used the theoretically justified $\alpha = 4$ for Algorithm 1. All code is available in the supplementary material. Each data-point is the average of 500 independent samples with the blue crosses and red squares indicating the regret of UCB-S and UCB respectively.

$K = 2, \mu_1(\theta) = \theta, \mu_2(\theta) = -\theta,$
$n = 50\,000$ (see Figure 2.(a))

$K = 2, \mu_1(\theta) = \theta, \mu_2(\theta) = -\theta,$
$\theta = 0.04$ (see Figure 2.(a))

$K = 2, \mu_1(\theta) = 0, \mu_2(\theta) = \theta,$
$n = 50\,000$ (see Figure 2.(b))

The results show that Algorithm 1 typically out-performs regular UCB. The exception is the top right experiment where UCB performs slightly better for $\theta < 0$. This is not surprising, since in this case the structured version of UCB cannot exploit the additional structure and suffers due to worse constant factors. On the other hand, if $\theta > 0$, then UCB endures logarithmic regret and performs significantly worse than its structured counterpart. The superiority of Algorithm 1 would be accentuated in the top left and bottom right experiments by increasing the horizon.

$K = 2, \mu_1(\theta) = \theta \mathbb{1}\{\theta > 0\},$
$\mu_2(\theta) = -\theta \mathbb{1}\{\theta < 0\},$
$n = 50\,000$ (see Figure 2.(c))

# 8 Conclusion

The limitation of the new approach is that the proof techniques and algorithm are most suited to the case where the number of actions is relatively small. Generalising the techniques to large action spaces is therefore an important open problem. There is still a small gap between the upper and lower bounds, and the lower bounds have only been proven for special examples. Proving a general problem-dependent lower bound is an interesting question, but probably extremely challenging given the flexibility of the setting. We are also curious to know if there exist problems for which the optimal regret is somewhere between finite and logarithmic. Another question is that of how to define Thompson sampling for structured bandits. Thompson sampling has recently attracted a great deal of attention [13, 2, 14, 3, 9], but so far we are unable even to define an algorithm resembling Thompson sampling for the general structured bandit problem. Not only because we have not endowed $\Theta$ with a topology, but also because choosing a reasonable prior seems rather problem-dependent. An advantage of our approach is that we do not rely on knowing the distribution of the rewards while with one notable exception [9] this is required for Thompson sampling.

**Acknowledgements.** Tor Lattimore was supported by the Google Australia Fellowship for Machine Learning and the Alberta Innovates Technology Futures, NSERC. The majority of this work was completed while Rémi Munos was visiting Microsoft Research, New England. This research was partially supported by the European Community's Seventh Framework Programme under grant agreements no. 270327 (project CompLACS).

## Footnotes

[1]Current affiliation: Google DeepMind.

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

# A  Ambiguous Case

We assume for convenience that $K = 2$ and $\mu_1(\theta) \neq \mu_2(\theta)$ for all $\theta \in \Theta$. The second assumption is non-restrictive, since an algorithm cannot perform badly on the $\theta$ for which $\mu_1(\theta) = \mu_2(\theta)$, so we can simply remove these points from the parameter space. Now $\Theta$ can be partitioned into three sets according to whether or not finite regret is expected by Theorem 3, or impossible by Theorem 9.

$$\Theta_{\text{easy}} := \left\{ \theta \in \Theta : \exists \varepsilon > 0 \text{ such that } \left| \mu_{i^*(\theta)}(\theta') - \mu_{i^*(\theta)}(\theta') \right| < \varepsilon \implies i^*(\theta') = i^*(\theta) \right\}$$

$$\Theta_{\text{hard}} := \left\{ \theta \in \Theta : \exists \theta' \in \Theta \text{ such that } \mu_{i^*(\theta)}(\theta) = \mu_{i^*(\theta)}(\theta') \text{ and } i^*(\theta') \neq i^*(\theta) \right\}$$

$$\Theta_{\text{amb}} := \Theta - \Theta_{\text{easy}} - \Theta_{\text{hard}}$$

The topic of this section is to study whether or not finite regret is possible on $\Theta_{\text{amb}}$, and what sacrifices need to be made in order to achieve this. Some examples are given in Figure 4. Note that (a) was considered by Bubeck et. al. [11, Thm. 8] and will receive special attention here.

**Figure 4:** Ambiguous examples

The main theorem is this section shows that finite regret is indeed possible for many $\theta \in \Theta_{\text{amb}}$ without incurring significant additional regret for $\theta \in \Theta_{\text{easy}}$ and retaining logarithmic regret for $\theta \in \Theta_{\text{hard}}$. The following algorithm is similar to Algorithm 1, but favours actions which may be optimal for some plausible ambiguous $\theta$. Theorems will be given subsequently, but proofs are omitted.

---

**Algorithm 2**

---

1: **Input:** functions $\mu_1, \cdots, \mu_k : \Theta \to [0, 1]$, $\{\beta_t\}_{t=1}^{\infty}$

2: $\kappa_1 = 0$

3: **for** $t \in 1, \ldots, \infty$ **do**

4:      Define confidence set: $\tilde{\Theta}_t \leftarrow \left\{ \tilde{\theta} : \forall i, \ \left| \mu_i(\tilde{\theta}) - \hat{\mu}_{i,T_i(t-1)} \right| < \sqrt{\dfrac{\alpha \sigma^2 \log t}{T_i(t-1)}} \right\}$

5:      **if** $\kappa_t = 0 \ \wedge \ \exists \tilde{\Theta}_t \cap \Theta_{\text{amb}} \neq \emptyset$ **then**

6:          Choose $\theta \in \tilde{\Theta}_t \cap \Theta_{\text{amb}}$ arbitrarily and set $\kappa_t = i^*(\theta)$

7:      Choose $I_t = \arg\max_k \sup_{\theta \in \tilde{\Theta}_t} \mu_k(\theta) + \mathbb{1}\{k = \kappa_t\} \sqrt{\dfrac{\beta_t \log t}{T_k(t-1)}}$

8:      $\kappa_{t+1} \leftarrow \mathbb{1}\{I_t = \kappa_t\}$

---

**Theorem 10.** *Suppose* $K = 2$, $\theta \in \Theta$ *and* $i^* = 1$ *and* $\beta_t = \log\log t$ *and* $\Delta := \mu_1(\theta) - \mu_2(\theta)$. *Then Algorithm 2 satisfies:*

1. $\limsup_{n \to \infty} \mathbb{E}_\theta R_n(\theta) / \log n < \infty$.

2. *If* $\theta \in \Theta_{\text{easy}}$, *then* $\mathbb{E}_\theta R_n(\theta) \in O(\frac{1}{\Delta}(\log\frac{1}{\Delta})(\log\log\frac{1}{\Delta}))$.

3. *If* $\theta$ *is such that*

$$\lim_{\delta \to 0} \sup_{\theta' : |\mu_1(\theta) - \mu_1(\theta')| < \delta} \frac{\mu_2(\theta') - \mu_1(\theta')}{|\mu_1(\theta) - \mu_1(\theta')|} < \infty, \tag{11}$$

     *then* $\lim_{n \to \infty} \mathbb{E}_\theta R_n(\theta) < \infty$.

**Remark 11.** The condition (11) not satisfied for $\theta \in \Theta_{\text{hard}}$, since in this case there exists some $\theta'$ with $\mu_1(\theta) = \mu_1(\theta')$ but where $\mu_2(\theta') - \mu_1(\theta') > 0$. The condition may not be satisfied even for $\theta \in \Theta_{\text{amb}}$. See, for example, Figure 4.(c). The condition *is* satisfied for all other ambiguous $\theta$ for the problems shown in Figure 4.(a,b,d) where the risk $\mu_2(\theta') - \mu_1(\theta')$ decreases linearly as $\mu_1(\theta) - \mu_1(\theta')$ converges to zero.

The following theorem shows that you cannot get finite regret for the ambiguous case where (11) is not satisfied without making sacrifices in the easy case.

**Theorem 12.** *Suppose* $\theta \in \Theta_{\text{amb}}$ *with* $i^*(\theta) = 1$ *and* $\mathbb{E}_\theta R_n(\theta) \in O(1)$. *Then there exists a constant* $c > 0$ *such that for each* $\theta'$ *with* $i^*(\theta') = 2$ *we have*

$$\mathbb{E}_{\theta'} R_n(\theta') \geq c \frac{\mu_2(\theta') - \mu_1(\theta')}{(\mu_1(\theta) - \mu_1(\theta'))^2}.$$

Therefore if the condition (3) in the statement of Theorem 10 is not satisfied for some ambiguous $\theta$, then we can construct a sequence $\{\theta'\}_{i=1}^{\infty}$ such that $\lim_{i \to \infty} \mu_1(\theta_i') = \mu_1(\theta)$ and where

$$\lim_{i \to \infty} (\mu_2(\theta_i') - \mu_1(\theta_i')) \mathbb{E}_{\theta_i'} R_n(\theta_i') = \infty,$$

which means that the regret must grow faster than the inverse of the gap. The situation becomes worse the faster the quantity below diverges to infinity.

$$\sup_{\theta':|\mu_1(\theta)-\mu_1(\theta')|<\delta} \frac{\mu_2(\theta') - \mu_1(\theta)}{|\mu_1(\theta) - \mu_1(\theta')|}.$$

In summary, finite regret is often possible in the ambiguous case, but may lead to worse regret guarantees in the easy case. Ultimately we are not sure how to optimise these trade-offs and there are still many interesting unanswered questions.

**Analysis of Figure 4.(a)**

We now consider a case of special interest that was previously studied by Bubeck et. al. [11] and is depicted in Figure 4.(a). The structured bandit falls into the ambiguous case when $\theta \leq 0$, since no interval about $\mu_1(\theta) = 0$ is sufficient to rule out the possibility that the second action is in fact optimal. Nevertheless, using a carefully crafted algorithm we show that the optimal regret is smaller than one might expect. The new algorithm operates in phases, choosing each action a certain number of times. If all evidence points to the first action being best, then this is taken until its optimality is proven to be implausible, while otherwise the second action is taken. The algorithm is heavily biased towards choosing the first action where estimation is more challenging, and where the cost of an error tends to be smaller.

**Algorithm 3**

---

1: $\alpha \leftarrow 5$
2: **for** $\ell \in 2, \ldots, \infty$ **do**    // Iterate over phases
3:    $n_{1,\ell} = 2^\ell$ and $n_{2,\ell} = \ell^2$
4:    Choose each arm $k \in \{1, 2\}$ exactly $n_{k,\ell}$ times and let $\hat{\mu}_{k,\ell,n_{k,\ell}}$ be the average return
5:    $s \leftarrow 0$
6:    **if** $\hat{\mu}_{1,\ell,n_{1,\ell}} \geq -\sqrt{\frac{\alpha}{n_{1,\ell}} \log \log n_{1,\ell}}$ and $\hat{\mu}_{2,n_{2,\ell}} < -1/2$ **then**
7:        **while** $\hat{\mu}_{1,\ell,n_{1,\ell}+s} \geq -\sqrt{\frac{\alpha \log \log(n_{1,\ell}+s)}{n_{1,\ell}+s}}$ **do**
8:            Choose action 1 and $s \leftarrow s + 1$ and $\hat{\mu}_{1,\ell,n_{1,\ell}+s}$ is average return of arm 1 this phase
9:    **else**
10:        **while** $\hat{\mu}_{2,\ell,n_{2,\ell}+s} \geq -\frac{1}{2}$ **do**
11:            Choose action 2 and $s \leftarrow s + 1$ and $\hat{\mu}_{2,\ell,n_{2,\ell}+s}$ is average return of arm 2 this phase

---

**Theorem 13.** *Let $\Theta = [-1, 1]$ and $\mu_1(0) = -\theta \mathbb{1}\{\theta > 0\}$ and $\mu_2(\theta) = -\mathbb{1}\{\theta \leq 0\}$. Assume returns are normally distributed with unit variance. Then Algorithm 3 suffers regret bounded by*

$$\mathbb{E}_\theta R_n(\theta) \in \begin{cases} O\left(\frac{1}{\theta} \log \log \frac{1}{\theta}\right) & \text{if } \theta > 0 \\ O(1) & \text{otherwise.} \end{cases}$$

**Remark 14.** Theorem 13 contradicts a result by Bubeck et. al. [11, Thm. 8], which states that for any algorithm

$$\max\left\{ \mathbb{E}_0 R_n(0), \sup_{\theta > 0} \theta \cdot \mathbb{E}_\theta R_n(\theta) \right\} \in \Omega\left(\log n\right).$$

But by Theorem 13 there exists an algorithm for which

$$\max\left\{ \mathbb{E}_0 R_n(0), \sup_{\theta > 0} \theta \cdot \mathbb{E}_\theta R_n(\theta) \right\} \in O\left( \sup_{\theta > 0} \min\left\{ \theta^2 n, \log \log \frac{1}{\theta} \right\} \right) = O(\log \log n).$$

We are currently unsure whether or not the dependence on $\log \log \frac{1}{\theta}$ can be dropped from the bound given in Theorem 13. Note that Theorem 3 cannot be applied when $\theta = 0$, so Algorithm 1 suffers logarithmic regret in this case. Algorithm 3 is carefully tuned and exploits the asymmetry in the problem. It is possible that the result of Bubeck et. al. can be saved in spirit by using the symmetric structured bandit depicted in Figure 4.(b). This would still only give a worst-case bound and does not imply that finite problem-dependent regret is impossible.

*Proof of Theorem 13.* It is enough to consider only $\theta \in [0, 1]$, since the returns on the arms is constant for $\theta \in [-1, 0]$. We let $L$ be the number phases (times that the outer loop is executed) and $T_\ell$ be the number of times the sub-optimal action is taken in the $\ell$th phase. Recall that $\hat{\mu}_{k,\ell,t}$ denotes the empirical estimate of $\mu$ based on $t$ samples taken in the $\ell$th phase.

**Step 1: Decomposing the regret**

The regret is decomposed:

$$(\theta = 0): \qquad \mathbb{E}_0 R_n(0) = \mathbb{E}_0 \sum_{\ell=0}^{L} T_\ell = \sum_{\ell=0}^{\infty} \mathbb{P}_0 \{L \geq \ell\} \, \mathbb{E}_0[T_\ell | L \geq \ell]$$

$$(\theta > 0): \qquad \mathbb{E}_\theta R_n(\theta) = \theta \mathbb{E}_\theta \sum_{\ell=0}^{L} T_\ell = \theta \sum_{\ell=0}^{\infty} \mathbb{P}_\theta \{L \geq \ell\} \, \mathbb{E}_\theta[T_\ell | L \geq \ell]$$

**Step 2: Bounding $\mathbb{E}_\theta[T_\ell | L \geq \ell]$**

We need to consider the cases when $\theta = 0$ and $\theta > 0$ separately. If $s \geq 1$, then

$$\mathbb{P}_0 \{T_\ell \geq n_{2,\ell} + s | L \geq \ell\} \overset{(a)}{\leq} \mathbb{P}_0 \left\{ \hat{\mu}_{2,\ell,n_{2,\ell}+s-1} \geq -\frac{1}{2} \right\} \overset{(b)}{=} \mathbb{P}_0 \left\{ \hat{\mu}_{2,\ell,n_{2,\ell}+s-1} - \mu_2(0) \geq \frac{1}{2} \right\}$$

$$\overset{(c)}{\leq} \exp\left( -\frac{1}{2}(n_{2,\ell} + s - 1) \right) \overset{(d)}{\leq} \exp\left( -\frac{s}{2} \right),$$

where (a) follows since if the second action is chosen more than $n_{2,\ell}$ times in the $\ell$th phase, then that phase ends when $\hat{\mu}_{2,\ell,t} < -\frac{1}{2}$, (b) by noting that $\mu_2(0) = -1$, (c) follows from the standard concentration inequality and the fact that unit variance is assumed, (d) since $n_{2,\ell} \geq 1$. Therefore by Lemma 17 we have that $\mathbb{E}_0[T_\ell | L \geq \ell] \leq n_{2,\ell} + 2e^{1/2}$. Now assume $\theta > 0$ and define

$$\omega_2(x) = \min\{z : y \geq x \log \log y, \; \forall y \geq z\},$$

which satisfies $\omega_2(x) \in O(x \log \log x)$. If $n_{1,\ell} + s - 1 \geq \omega_2\left(\frac{4\alpha}{\theta^2}\right)$, then

$$\mathbb{P}_\theta \{T_\ell \geq n_{1,\ell} + s | L \geq \ell\} \overset{(a)}{\leq} \mathbb{P}_\theta \left\{ \hat{\mu}_{1,\ell,n_{1,\ell}+s-1} \geq -\sqrt{\frac{\alpha}{n_{\ell,1}+s-1} \log \log(n_{1,\ell}+s-1)} \right\}$$

$$\overset{(b)}{=} \mathbb{P}_\theta \left\{ \hat{\mu}_{1,\ell,n_{1,\ell}+s-1} - \mu_1(\theta) \geq \theta - \sqrt{\frac{\alpha}{n_{\ell,1}+s-1} \log \log(n_{1,\ell}+s-1)} \right\}$$

$$\overset{(c)}{\leq} \mathbb{P}_\theta \left\{ \hat{\mu}_{1,\ell,n_{1,\ell}+s-1} - \mu_1(\theta) \geq \theta/2 \right\} \overset{(d)}{\leq} \exp\left( -\frac{\theta^2}{8}(n_{1,\ell}+s-1) \right) \overset{(e)}{\leq} \exp\left( -\frac{\theta^2}{8}s \right),$$

where (a) follows since if the first arm (which is now sub-optimal) is chosen more than $n_{1,\ell}$ times, then the phase ends if $\hat{\mu}_{1,\ell,t}$ drops below the confidence interval. (b) since $\mu_1(\theta) = -\theta$. (c) since $n_{1,\ell} + s - 1 \geq \omega_2\left(\frac{4\alpha}{\theta^2}\right)$. (d) by the usual concentration inequality and (e) since $n_{1,\ell} \geq 1$. Another application of Lemma 17 yields

$$\mathbb{E}_\theta[T_\ell | L \geq \ell] \leq \max\left\{ n_{1,\ell}, \; \omega_2\left(\frac{4\alpha}{\theta^2}\right) \right\} + \frac{8e^{\theta^2/8}}{\theta^2},$$

where the max appears because we demanded that $n_{1,\ell} + s - 1 \geq \omega_2\left(\frac{4\alpha}{\theta^2}\right)$ and since at the start of each phase the first action is taken at least $n_{1,\ell}$ times before the phase can end.

**Bounding the number of phases**

Again we consider the cases when $\theta = 0$ and $\theta \geq 0$ separately.

$$\mathbb{P}_0 \{L > \ell\} \overset{(a)}{\leq} \mathbb{P}_0 \left\{ \hat{\mu}_{2,\ell,n_{2,\ell}} \geq -\frac{1}{2} \vee \exists s : \hat{\mu}_{1,\ell,n_{1,\ell}+s} \leq -\sqrt{\frac{\alpha}{n_{1,\ell}+s} \log \log(n_{1,\ell}+s)} \right\}$$

$$\overset{(b)}{\leq} \mathbb{P}_0 \left\{ \hat{\mu}_{2,\ell,n_{2,\ell}} \geq -\frac{1}{2} \right\} + \mathbb{P}_0 \left\{ \exists s : \hat{\mu}_{1,\ell,n_{1,\ell}+s} \leq -\sqrt{\frac{\alpha}{n_{1,\ell}+s} \log \log(n_{1,\ell}+s)} \right\}$$

$$\overset{(c)}{\leq} \exp\left( -\frac{n_{2,\ell}}{8} \right) + \mathbb{P}_0 \left\{ \exists s : \hat{\mu}_{1,\ell,n_{1,\ell}+s} \leq -\sqrt{\frac{\alpha}{n_{1,\ell}+s} \log \log(n_{1,\ell}+s)} \right\}, \qquad (12)$$

where (a) is true since the $\ell$th phase will not end if $\hat{\mu}_{2,\ell,n_{2,\ell}} < -1/2$ and if $\hat{\mu}_{1,\ell,t}$ never drops below the confidence interval. (b) follows from the union bound and (c) by the concentration inequality. The second term is bounded using the maximal inequality and the peeling technique.

$$
\mathbb{P}_0 \left\{ \exists s : \hat{\mu}_{1,\ell,n_{1,\ell}+s} \leq -\sqrt{\frac{\alpha}{n_{1,\ell}+s} \log\log(n_{1,\ell}+s)} \right\}
$$

$$
\overset{(a)}{\leq} \sum_{k=0}^{\infty} \mathbb{P}_0 \left\{ \exists t : 2^k n_{1,\ell} \leq t \leq 2^{k+1} n_{1,\ell} \wedge \hat{\mu}_{1,\ell,t} \leq -\sqrt{\frac{\alpha}{t} \log\log t} \right\}
$$

$$
\overset{(b)}{\leq} \sum_{k=0}^{\infty} \mathbb{P}_0 \left\{ \exists t \leq 2^{k+1} n_{1,\ell} : \hat{\mu}_{1,\ell,t} \leq -\sqrt{\frac{\alpha}{n_{1,\ell}2^k} \log\log 2^k n_{1,\ell}} \right\}
$$

$$
\overset{(c)}{\leq} \sum_{k=0}^{\infty} \exp\left(-\alpha \log\log\left(2^k n_{1,\ell}\right)\right) \overset{(d)}{=} \sum_{k=0}^{\infty} \left(\frac{1}{\log 2^k + \log n_{1,\ell}}\right)^{\alpha} \overset{(e)}{\leq} \frac{2}{\log 2} \left(\frac{1}{\ell \log 2}\right)^{\alpha-1}
$$

where (a) follows by the union bound, (b) by bounding $t$ in the interval $2^k n_{1,\ell} \leq t \leq 2^{k+1} n_{1,\ell}$. (c) follows from the maximal inequality. (d) is trivial while (e) follows by approximating the sum by an integral. By combining with (12) we obtain

$$
\mathbb{P}_0\{L > \ell\} \leq \exp\left(-\frac{n_{2,\ell}}{8}\right) + \frac{2}{\log 2}\left(\frac{1}{\ell \log 2}\right)^{\alpha-1} = \exp\left(-\frac{\ell}{8}\right) + \frac{2}{\log 2}\left(\frac{1}{\ell \log 2}\right)^{\alpha-1}.
$$

More straight-forwardly, if $\theta > 0$, then

$$
\mathbb{P}_\theta\{L > \ell\} \leq \mathbb{P}_\theta\left\{\exists s : \hat{\mu}_{2,n_{2,\ell}+s} < -\frac{1}{2}\right\} \leq 5\exp\left(-\frac{n_{2,\ell}}{16}\right),
$$

where in the last inequality we used Lemma 16 and naive bounding.

**Putting it together**

We now combine the results of the previous components to obtain the required bound on the regret. Recall that $\alpha = 5$.

$$
(\theta = 0): \quad \mathbb{E}_0 R_n(0) = \mathbb{E}_\theta \sum_{\ell=2}^{\infty} T_\ell = \sum_{\ell=2}^{\infty} \mathbb{P}_0\{L \geq \ell\} \mathbb{E}_0[T_\ell | L \geq \ell]
$$

$$
\leq \sum_{\ell=2}^{\infty} \left(\exp\left(-\frac{n_{2,\ell}}{8}\right) + \frac{2}{\log 2}\left(\frac{1}{\ell \log 2}\right)^{\alpha-1}\right)\left(n_{2,\ell} + 2e^{1/2}\right)
$$

$$
= \sum_{\ell=2}^{\infty} \left(\exp\left(-\frac{\ell}{8}\right) + \frac{2}{\log 2}\left(\frac{1}{\ell \log 2}\right)^{\alpha-1}\right)\left(\ell^2 + 2e^{1/2}\right) \in O(1)
$$

$$
(\theta > 0): \quad \mathbb{E}_\theta R_n(\theta) = \theta\mathbb{E}_\theta \sum_{\ell=2}^{\infty} T_\ell = \theta \sum_{\ell=1}^{\infty} \mathbb{P}\{L \geq \ell\} \mathbb{E}[T_\ell | L \geq \ell]
$$

$$
\leq 5\theta \sum_{\ell=2}^{\infty} \exp\left(-\frac{n_{2,\ell}}{16}\right)\left(\max\left\{n_{1,\ell}, \omega_2\left(\frac{4\alpha}{\theta^2}\right)\right\} + \frac{8e^{\theta^2/8}}{\theta^2}\right)
$$

$$
\in O\left(\theta \cdot \omega_2\left(\frac{\alpha}{\theta^2}\right)\right) = O\left(\frac{1}{\theta}\log\log\frac{1}{\theta}\right). \qquad \square
$$

## B   Technical Lemmas

**Lemma 15.** *Define functions $\omega$ and $\omega_2$ by*

$$
\omega(x) := \min\{z > 1 : y \geq x \log y, \ \forall y \geq z\}
$$

$$
\omega_2(x) := \min\{z > e : y \geq x \log\log y, \ \forall y \geq z\}.
$$

*Then $\omega(x) \in O(x \log x)$ and $\omega_2(x) \in O(x \log\log x)$.*

**Lemma 16.** *Let $\{X_i\}_{i=1}^{\infty}$ be sampled from some sub-gaussian distributed arm with mean $\mu$ and unit sub-gaussian constant. Define $\hat{\mu}_t = \frac{1}{t}\sum_{s=1}^{t} X_s$. Then for $s \geq 6/\Delta^2$ we have*

$$\mathbb{P}\left\{\exists t \geq s : \hat{\mu}_t - \mu \geq \Delta\right\} \leq p + \frac{1}{\log 2}\log\frac{1}{1-p}$$

*where $p = \exp\left(-\frac{s\Delta^2}{4}\right)$.*

*Proof.* We assume without loss of generality that $\mu = 0$ and use a peeling argument combined with Azuma's maximal inequality

$$
\begin{aligned}
\mathbb{P}\left\{\exists t > s : \hat{\mu}_t \geq \Delta\right\} &\overset{(a)}{=} \mathbb{P}\left\{\exists k \in \mathbb{N}, 2^k s \leq t < 2^{k+1}s : \hat{\mu}_t \geq \Delta\right\} \\
&\overset{(b)}{=} \mathbb{P}\left\{\exists k \in \mathbb{N}, 2^k s \leq t < 2^{k+1}s : t\hat{\mu}_t \geq t\Delta\right\} \\
&\overset{(c)}{\leq} \mathbb{P}\left\{\exists k \in \mathbb{N}, 2^k s \leq t < 2^{k+1}s : t\hat{\mu}_t \geq 2^k s\Delta\right\} \\
&\overset{(d)}{\leq} \sum_{k=0}^{\infty}\mathbb{P}\left\{\exists 2^k s \leq t < 2^{k+1}s : t\hat{\mu}_t \geq 2^k s\Delta/2\right\} \\
&\overset{(e)}{\leq} \sum_{k=0}^{\infty}\mathbb{P}\left\{\exists t < 2^{k+1}s : t\hat{\mu}_t \geq 2^k s\Delta/2\right\} \\
&\overset{(f)}{\leq} \sum_{k=0}^{\infty}\exp\left(-\frac{1}{2}\frac{\left(2^k s\Delta\right)^2}{2^{k+1}s}\right) \overset{(g)}{=} \sum_{k=0}^{\infty}\exp\left(-\frac{2^k s\Delta^2}{4}\right) \\
&\overset{(h)}{=} \sum_{k=0}^{\infty}\exp\left(-\frac{s\Delta^2}{4}\right)^{2^k} \overset{(i)}{\leq} p + \frac{1}{\log 2}\log\frac{1}{1-p}
\end{aligned}
$$

where (a) follows by splitting the sum over an exponential grid. (b) by comparing cumulative differences rather than the means. (c) since $t > 2^k s$. (d) by the union bound over all $k$. (e) follows by increasing the range. (f) by Azuma's maximal inequality. (g) and (h) are true by straight-forward arithmetic while (i) follows from Lemma 18. $\qquad\square$

**Lemma 17.** *Suppose $z$ is a positive random variable and for some $\alpha > 0$ it holds for all natural numbers $k$ that $\mathbb{P}\left\{z \geq k\right\} \leq \exp(-k\alpha)$. Then $\mathbb{E}z \leq \frac{e^\alpha}{\alpha}$*

*Proof.* Let $\delta \in (0,1)$. Then

$$\mathbb{P}\left\{z \geq \log\frac{1}{\delta}\right\} \leq \mathbb{P}\left\{z \geq \left\lfloor\log\frac{1}{\delta}\right\rfloor\right\} \leq \exp\left(-\alpha\left\lfloor\log\frac{1}{\delta}\right\rfloor\right)$$

$$\leq e^\alpha \cdot \exp\left(-\alpha\log\left(\frac{1}{\delta}\right)\right) = e^\alpha \cdot \delta^\alpha.$$

To complete the proof we use a standard identity to bound the expectation

$$\mathbb{E}z \leq \int_0^1 \frac{1}{\delta}\mathbb{P}\left\{z \geq \log\frac{1}{\delta}\right\}d\delta \leq e^\alpha\int_0^1 \delta^{\alpha-1}d\delta = \frac{e^\alpha}{\alpha}. \qquad\square$$

**Lemma 18.** *Let $p \in (0, 7/10)$. Then $\displaystyle\sum_{k=0}^{\infty} p^{2^k} \leq p + \frac{1}{\log 2}\log\frac{1}{1-p}$.*

*Proof.* Splitting the sum and comparing to an integral yields:

$$
\begin{aligned}
\sum_{k=0}^{\infty} p^{2^k} &\overset{(a)}{=} p + \sum_{k=1}^{\infty} p^{2^k} \overset{(b)}{\leq} p + \int_0^{\infty} p^{2^k}dk \overset{(c)}{=} p + \frac{1}{\log 2}\int_1^{\infty} p^u/u\, du \\
&\overset{(d)}{\leq} p + \frac{1}{\log 2}\sum_{u=1}^{\infty} p^u/u \overset{(e)}{=} p + \frac{1}{\log 2}\log\frac{1}{1-p}
\end{aligned}
$$

where (a) follows by splitting the sum. (b) by noting that $p^{2^k}$ is monotone decreasing and comparing to an integral. (c) by substituting $u = 2^k$. (d) by reverting back to a sum. (e) follows from a standard formula. $\qquad\square$

## C   Table of Notation

| | |
|---|---|
| $K$ | number of arms |
| $\Theta$ | parameter space |
| $\theta^*$ | unknown parameter $\theta^* \in \Theta$ |
| $I_t$ | arm played at time-step $t$ |
| $T_i(n)$ | number times arm $i$ has been played after time-step $n$ |
| $X_{i,s}$ | $s$th reward obtained when playing arm $i$ |
| $\Delta_i$ | gap between the means of the best arm and the $i$th arm |
| $\Delta_{\min}$ | minimum gap, $\Delta_{\min} := \min_{i:\Delta_i>0} \Delta_i$ |
| $\Delta_{\max}$ | maximum gap, $\Delta_{\max} := \max_i \Delta_i$ |
| $A$ | set of arms $A := \{1, 2, \cdots, K\}$ |
| $A'$ | set of suboptimal arms $A := \{i : \Delta_i > 0\}$ |
| $R_n$ | regret at time-step $n$ given unknown true parameter $\theta^*$ |
| $R_n(\theta)$ | regret at time-step $n$ given parameter $\theta$ |
| $\mu_i(\theta)$ | mean of arm $i$ given $\theta$ |
| $\hat{\mu}_{i,s}$ | empiric estimate of the mean of arm $i$ after $s$ plays |
| $\mu^*(\theta)$ | maximum return at $\theta$. $\mu^*(\theta) := \max_i \mu_i(\theta)$ |
| $i^*$ | optimal arm given $\theta^*$ |
| $i^*(\theta)$ | optimal arm given $\theta$ |
| $\omega(x)$ | minimum value $y$ such that $z \geq x \log z$ for all $z \geq y$ |
| $\omega_2(x)$ | minimum value $y$ such that $z \geq x \log \log z$ for all $z \geq y$ |
| $F_t$ | event that the true value of some mean is outside the confidence interval about the empiric estimate at time-step $t$ |
| $\alpha$ | parameter controlling how exploring the algorithm UCB-S is |
| $\sigma^2$ | known parameter controlling the tails of the distributions governing the return of the arms |
| $u_i(n)$ | critical number of samples for arm $i$. $u_i(n) := \left\lceil \frac{8\sigma^2 \alpha \log n}{\Delta_i^2} \right\rceil$ |