[Reviews · NeurIPS 2014]

Submitted by Assigned_Reviewer_8

Paper Summary:

This paper treats a general multi-armed bandit problem in which the mean reward of each arm depends on a common unknown parameter. The authors consider a simple modification of the UCB1 algorithm. They show, unsurprisingly, that the algorithm satisfies a regret bound like that of UCB1. The main improvement of this paper is to show when the optimal arm can be identified perfectly by samples of the optimal arm, algorithm’s regret is bounded by a constant independent of the time horizon.

Overall:

The bounds provided are pretty strong, and the proofs in the body of the paper are clear and well organized. The explicit finite time analysis here is novel. My main issues with the paper are (1) it's already known in the literature exactly when finite regret is possible and (2) I worry that the algorithm that is considered is not appropriate for more practical structured bandit problems. If the paper is accepted, I think that these two points need to be acknowledged clearly.

Quality and Clarity:

This paper is well executed. The bounds provided are pretty strong, and the proofs in the body of the paper are clear and well organized. A few minor typos are listed at the bottom of this review.

Originality

[1] showed that finite regret is possible under the same condition as in this work, but otherwise regret must grow logarithmically with the time horizon. That paper is focused on providing an asymptotically exact characterization of regret and makes the technical assumption that the parameter space is finite. It’s mostly dismissed by the authors for this reason. But Graves and Lai later removed the condition that the parameter space is finite. There are also four existing papers on permutation bandits, a special case under which finite regret is possible. In addition, the algorithm considered seems to have also been studied before, for example in [Mailard and Mannor; Latent Bandits; 2014].

The main original contributions are (1) to show that a simple modification of UCB is enough to attain finite regret, and (2) to provide an explicit finite time analysis of how large this finite regret may be.

-Todd L Graves and Tze Leung Lai. Asymptotically eļ¬ƒcient adaptive choice of control laws in controlled markov chains. SIAM journal on control and optimization, 35(3):715–743, 1997

Significance:

I worry that the improvements are mostly relevant for some very simple structured bandit problems, and the algorithm considered is not appropriate for more practical structured bandit problems.

The proposed algorithm is a very simple modification of UCB1. It maintains independent confidence intervals for each arm, and constructs a confidence set containing the parameters consistent with these intervals. This is nice, because it’s interesting that a simple modification of UCB is enough to attain finite regret. On the other hand, the proposed algorithm is in some ways a major step backward relative other UCB-style algorithms for problems with dependent arms. Consider, for example, a linear bandit problem. Even if no single arm were sampled many times, the confidence sets of (e.g. [Dani et. al; Stochastic linear optimization under bandit feedback; 2008]) could shrink around the true parameter because there is a known relationship among the arms. This is also true of algorithms in [10,18]. The confidence sets in this paper don’t fully exploit the known structure of the problem and therefore don’t share this property.

Minor comments:
-Lines 35-38: “This model has already been considered when the dependencies are linear by [17], but not as far as we are aware for the general non-linear case.” You actually cite two papers [1, 18] that consider the general non-linear case, so this is a strange comment. There is also a paper by Graves and Lai, and a Thompson sampling paper by Gopalan, Mannor and Mansour in ICML this year.
- The motivating example in lines 45-49 is a strange choice given the preceding pitch, since arms are modeled independently in this example.
- You sometimes slip up and write the true parameter as \theta instead of \theta^*. See for example Theorem 2 and its proof, or the proof of Lemma 5.
- Similarly, you sometimes slip up in your notation and use \mu _i to denote the true mean reward of arm i when really this should be denoted by \mu_i(\theta^*). See e.g. lines 92-93.
-In remark 4 you refer to a regime where n is large, but the preceding bound has no dependence on n.
- This is a picky comment… in lines 240-241, you don’t assume concentration inequalities. You assume rewards are sub-Gaussian, which implies concentration inequalities that you can then reference in this step.
- When referencing Lemma 5, you often use the expression given after equality ( e), so this should be in the statement of the result and not just the proof.
- Line 272, equality (b) is an inequality.
- In the proof of Lemma 7, given the current problem formulation it seems like theta-tilde and \theta are completely arbitrary. I believe you mean to specify that theta-tilde is in the confidence set, and to write theta^* instead of theta.
-In line 324, the “if” after (b) is a typo.

****************************************
Post-rebuttal comments:
****************************************

This is a well executed paper that makes several technical contributions. I think it should be accepted.

Let me apologize. The Lai + Graves paper doesn't seem to explicitly state that regret is finite, only that its growth rate is sub-logarithmic. So regret could grow as log(log(t)) for example.

Still, the authors should clarify that the condition for finite regret -- "choosing the optimal arm is sufficient to prove that it is indeed optimal" -- plays a very important role in past work. Were this a fundamentally new insight, I would have given the paper a very enthusiastic review.
Summary: The results here are solid, but in light of existing literature I'm not sure whether the contribution is significant enough. I am hesitating between a 5 and 6.

Submitted by Assigned_Reviewer_19

A bounded regret for a general class of parameterized multi-armed bandits is proven, under the assumption that the optimal arm in a neighborhood of true parameter remains the same. Also some negative result which shows that in some examples this bound is tight have been presented. The results improve on the standard bounds of multi-armed bandit by a factor of Log(T). Though this is not the first finite regret bound for the structured bandit it arguably applies to a more general class of problems than the prior work.

The results are fairly interesting and the contribution seems to be enough for a conference paper. Also the technical proofs seem to be correct and well-explained. I think the results in most part are original though the proposed algorithm has been already introduced in other papers (see item 1) but for different setting. Also as it is recognized by the authors the result of parametrized bandits of Agrawel leads to a similar bounded regret although for a more restrictive setting.

A few comments

1- Algorithm 1 is claimed to be novel. This seems not to be an entirely a correct statement. In fact rather similar algorithms have been already introduced and analyzed e.g., in latent bandits of (Maillard 14) and in transfer bandits of (Azar 13). Although those paper consider a slightly different setting but the algorithm is almost identical. These prior works need to be covered.

2- Some notation like \omega(x) and \omega^* as well as non-dominant terms of the bound can be simplified and better explained.

-----
Post-rebuttal: I think the authors have clearly addressed the main concerns of the reviewers in their author feedback especially concerning the issue of comparison with the prior work and the contribution w.r.t the state-of-the-art. I think, after some minor revisions, the paper is ready for publication.
Summary: Interesting results and fairly general setting. Rigorous analysis and well written.

Submitted by Assigned_Reviewer_39

Pre-rebuttal comments:

*********************************"******

In this paper, the authors study stochastic bandit problem with generalization (i.e. "the expected return of one arm may depend on the returns of other arms"). The authors have proposed a UCB-based algorithm for this problem, and proved that under certain conditions this algorithm is possible to achieve finite cumulative regret. The authors also show that the proposed algorithm is near-optimal in some special cases by deriving lower bounds in such cases.

The proposed algorithm (UCB-S) is quite straightforward given existing literature on bandit, and the problem of attaining O(1) regret in bandit is not new. Moreover, the identified condition ("if the samples from the optimal arm are sufficient to learn the optimal action") for O(1) regret is also not new (see Agrawal (1989) and Graves and Lai (1997)). It seems to me that the major contribution of this paper is that the authors have shown that under a known condition, a straightforward modification of UCB1 achieves O(1) regret. Furthermore, the derived O(1) bound is explicit. This result is interesting, and might be marginally above the acceptance threshold.

The analysis in this paper looks technically correct. The paper is well-written in general, but I recommend the authors to rewrite Section 6 to make it more readable.

Some minor comments:

(1) \theta in the proof of Lemma 5 and the proof of Lemma 7 should be \theta^\ast, based on the notation definition in Section 2.

(2) equality in Eqn(5) should be \leq.

(3) expectation on the left-hand side of Eqn(6) is missing.

One question: does the conclusion in Theorem 9 hold for all algorithms, as in Theorem 8?

Post-rebuttal comments:
*********************************************

After going through the authors' rebuttal, I think the authors have done a good job of comparing their paper with Graves and Lai (1997), and other relevant literature. I think the authors have addressed my concerns about the novelty of the paper, and the paper is ready to be published in NIPS after revision.
Summary: The paper is interesting, and I think it is ready to be published in NIPS after revision.
Author Feedback
Author rebuttal: Thank you to all reviewers for your time. We agree with the reviewers that note the proposed algorithm is not
entirely novel in the sense that roughly equivalent variants have been used in a variety of different settings, including
those pointed out by reviewers. Relevant references and discussion will be added.

Two reviewers mentioned the paper by Graves and Lai, which is certainly relevant and to which we
will include a reference and discussion. We would like to emphasize some differences between our
work and theirs:

1. They are shooting for an asymptotically optimal algorithm, so naturally focus on the case
where logarithmic regret is the best possible. Actually we could not find a claim that finite
regret occurs if B(\theta) is empty (Definition in Eq 2.4). This would seem to be implied by Eq 3.14,
but actually this only gives that the regret is sub-logarithmic (see the formal claim, Theorem 2). Maybe
following the proof carefully would lead to a finite regret, but this is not clear to us. On the other
hand, we emphasize the case where bounded regret is possible, and the dependence of this regret on the problem.

2. The results depend on the parameter space being a metric space and the optimal policy being constant locally
about the true parameter. For our work we make no assumptions on the parameter space.

3. (as pointed out by reviewers) they do not give an explicit computation of the regret bound in the finite case.

4. Lai and Graves give no meaningful lower bounds for the case where finite regret is possible, while we have lower bounds that
nearly match the upper bound in a number of important cases.

Thus the previous work that is most relevant to ours is by Bubeck et. al. [1,2] in which explicit finite
regret bounds are provided. However they obtain finite regret only in the case where the mean of the best arm and
a lower bound on the minimum gap are known. We extend those results and provide explicit finite regret
bounds in more general settings.

******************************************************
Comments specifically for reviewer 1:
******************************************************

Thank you for your comments, additional references and suggestions for improvement.

******************************************************
Comments specifically for reviewer 2:
******************************************************

Theorem 9 does indeed apply to all algorithms. The point here (as remarked after the theorem statement) is that the
new algorithm is making the right trade-off for this kind of problem. I.e, if sub-linear regret is expected for all
parameters (seems desirable), then no algorithm can avoid logarithmic regret in the natural type of problem
covered by the theorem.

We would like to point out that as well as explicitly deriving upper and lower the bounds on the regret in the finite
case, we also consider the ambiguous setting where knowing the exact value of the optimal arm is good enough to know that it is
optimal, but no imprecise estimate is sufficient (discussed in Section 6 and the Appendix). This case is not considered by Lai
and Graves, since their continuity assumption eliminates this possibility. Additionally we also find errors in previous work
by Bubeck et. al. [1], and prove (non-trivial) that the claimed result (Theorem 8 in [1]) cannot be fixed.

******************************************************
Comments specifically for reviewer 3:
******************************************************

Thank your for the careful review. If accepted we will address all minor comments, but also especially the
two points raised in your "Overall:" summary.

See also the second comment to reviewer 2 where we point out some additional contributions.

[1] Sebastien Bubeck, Vianney Perchet and Philippe Rigollet. Bounded regret in stochastic multi-armed bandits, COLT 2013

[2] Sebastien Bubeck and Che-Yu Liu. Prior-free and prior-dependent regret bounds for Thompson Sampling, NIPS 2013